# Sodium Content in Pre-Packaged Foods in China: A Food Label Analysis

**DOI:** 10.3390/nu15234862

**Published:** 2023-11-21

**Authors:** Xin Ding, Wanting Lv, Yang Liu, Jiewei Long, Hanning Li, Aiguo Ma, Yuexin Yang, Zhu Wang, Chao Gao

**Affiliations:** 1Institute of Nutrition and Health, School of Public Health, Qingdao University, Qingdao 266071, China; 2021021102@qdu.edu.cn (X.D.);; 2National Institute for Nutrition and Health, Chinese Center for Disease Control and Prevention, Beijing 100050, China; 3Department of Nutrition and Food Hygiene, School of Public Health, Tongji Medical College, Huazhong University of Science and Technology, Wuhan 430030, China; m202175527@hust.edu.cn; 4Chinese Nutrition Society, Beijing 100053, China; 5Laboratory of Trace Element Nutrition of National Health Commission, Beijing 100050, China

**Keywords:** sodium, sodium reduction, pre-packaged foods, nutrition labelling

## Abstract

Sodium intake from pre-packaged foods is increasing in China and is well above the WHO recommendation of 5 g per day. The purpose of this study is to analyze the sodium content of pre-packaged foods collected by the National Institute for Nutrition and Health, Chinese Center for Disease Control and Prevention (NINH, China CDC) in 20 provinces of China from 2017 to 2022. The proportion of pre-packaged foods that meet or exceed the low-sodium, medium-sodium, and high-sodium classifications were analyzed. The proportion of pre-packaged foods that meet and do not meet the WHO global sodium benchmarks and the difference in sodium content between these foods was also calculated. High-sodium foods include sauces, dips, and dressings (3896 mg/100 g), convenience foods (1578 mg/100 g), processed fish products (1470 mg/100 g), processed meat products (1323 mg/100 g), processed poultry products (1240 mg/100 g), snack foods (750 mg/100 g), processed egg products (741 mg/100 g), and fine dried noodles (602 mg/100 g). A large number of pre-packaged foods currently collected in China have a sodium content above sodium benchmarks. This study provided data to support the assessment of sodium intake from pre-packaged foods in the Chinese population and the implementation of comprehensive salt reduction strategies.

## 1. Introduction

High sodium intake is an important risk factor for high blood pressure [1,2], cardiovascular disease [3], stroke [4], stomach cancer [5], and premature mortality [6]. High sodium intake has led to 1.9 million deaths globally [7]. The World Health Organization (WHO) recommends a sodium intake for adults of less than 2 g/day (i.e., 5 g/day of salt) [8]. However, the current Chinese adult’s average salt intake is estimated to be 9.3 g/day per person [9], which is 86% higher than the WHO’s recommendation [8]. The consumption of pre-packaged foods in China continues to increase [10,11].

WHO makes sodium reduction a global priority and “best buy” [12,13]. In 96 national salt reduction initiatives, 89 countries combined two or more implementation strategies, including interventions in settings, food reformulation, consumer education, front-of-pack labelling (FOPL), and salt taxation [14]. The UK [15], the USA [16], and Canada [17] set voluntary salt reduction targets for pre-packaged foods; the South African government set maximum sodium content in various categories of processed foods [18]; Chile, Peru, and Uruguay adopted a front-of-pack warning label (FOPWL) [19,20]; and Fiji and Mexico adopted taxes on foods high in salt [14].

China implemented the Standard on Nutrition Labelling of Pre-packaged Foods (GB 28050–2011) in 2013 and made it mandatory to label the sodium content of pre-packaged foods [21]. The Healthy China 2030 Plan, the Healthy China Initiative (2019–2030), and the National Nutrition Plan (2017–2030) are the signature national Chinese population health policies that clarify specific measures to reduce salt, including promoting salt reduction in pre-packaged foods, revising the Standard on Nutrition Labelling of Pre-packaged Foods, and promoting the establishment of a food nutrition standards system. The National Institute for Nutrition and Health, Chinese Center for Disease Control and Prevention (NINH, China CDC) and the Chinese Nutrition Society jointly issued Guidelines for Salt Reduction in Chinese Food Industry in 2019 and proposed targets for 2025 and 2030 for the phased reduction of salt in pre-packaged foods [22].

Understanding the sodium content of pre-packaged foods on nutrition labels is essential for implementing comprehensive salt reduction strategies. Currently, there are insufficient data on the sodium content of pre-packaged foods in China, so the purpose of this study is to analyze the sodium content of pre-packaged foods collected by NINH, China CDC in 20 provinces of China from 2017 to 2022. And the proportion of pre-packaged foods that meet or exceed the low-sodium, medium-sodium, and high-sodium were analyzed. We also calculated the proportion of pre-packaged foods that meet and do not meet the WHO global sodium benchmarks, as well as the difference in sodium content of the corresponding foods. This study provided data to support the assessment of sodium intake from pre-packaged foods in the Chinese population and the implementation of comprehensive salt reduction strategies to help reduce sodium intake among the Chinese population and further reduce diet-related noncommunicable diseases (NCDs).

## 2. Materials and Methods

### 2.1. Data Collection for Chinese Pre-Packaged Foods

NINH, China CDC collected information on nutrition labelling of major pre-packaged foods from 20 provinces in 2017–2022. The collection provinces cover seven major geographical regions of China: Northeast China (Heilongjiang Province, Jilin Province, Liaoning Province); East China (Jiangsu Province, Zhejiang Province, Jiangxi Province, Shandong Province); North China (Beijing, Hebei Province, Inner Mongolia Autonomous Region); Central China (Henan Province); South China (Guangdong Province, Hainan Province); Southwest China (Guizhou Province, Chongqing Municipality, Xizang Autonomous Region); and Northwest China (Gansu Province, Qinghai Province, Ningxia Hui Autonomous Region, Xinjiang Uygur Autonomous Region). These regions include regions with high population density and economic correlation in China (Guangdong Province, Jiangsu Province, Shandong Province, Zhejiang Province, Henan Province, Hebei Province, Beijing, Chongqing, Inner Mongolia Autonomous Region) and regions with low population density and economic correlation (Xizang Autonomous Region, Qinghai Province, Hainan Province, Gansu Province, Jilin Province, Heilongjiang Province, Xinjiang Uygur Autonomous Region, Guizhou Province).

The staff photographed all sides of the pre-packaged food and captured all information from the product packaging (e.g., product name, brand, nutrition information panels (NIPs), and ingredients) and uploaded this information into the China Standardized Database for the Composition of Pre-packaged Food. The principle of collecting pre-packaged foods is that the pre-packaged food is within the shelf life, the food information on the pre-packaged food is clear, and the total energy converted from fat, carbohydrates, and protein on the label does not exceed the energy value on the label. Nutrient content in products was standardized to volumes per 100 g or 100 mL by the staff of NINH, China CDC.

### 2.2. Data Classification

According to the Standard on Nutrition Labelling of Pre-packaged Foods (GB 28050-2011) [21], the Standards for Uses of Food Additive (GB 2760-2014) [23], Regulation of Food Composition Data Expression (WS/T 464-2015) [24], as well as various national standards, group standards, and industry standards for food published by the Chinese government (Appendix A), with full consideration of the characteristics of pre-packaged foods, their processing technologies, formulation, and Chinese dietary habits, pre-packaged foods have been classified step by step. The specific number and distribution of various pre-packaged foods are shown in Table 1 and Figure 1.

### 2.3. Data Inclusion and Exclusion Criteria

Pre-packaged foods with complete NIPs and ingredients were included. The Standard on Nutrition Labelling of Pre-packaged Foods (GB 28050-2011) specifies mandatory rules for nutrition labelling by manufacturers to provide quantitative information on energy, protein, fat, carbohydrate, and sodium content of foods and their contributions to the Nutrient Reference Value (NRV). Foods with incorrect or incomplete nutrition information and foods that were missing ingredient information or could not be classified into target categories were excluded from the analysis.

### 2.4. Data Analysis

Data for sodium (mg/100 g) for each subcategory of pre-packaged foods are presented as median, proportion, range, and IQR means interquartile range.

According to the Standard on Nutrition Labelling of Pre-packaged Foods (GB 28050-2011), foods with a sodium content not higher than 120 mg/100 g were classified into low-sodium foods [21]. According to the recommendation of Dietary Guidelines for Chinese (2022), sodium content that exceeds 30% of NRV is high; thus, foods with sodium content higher than 600 mg/100 g were classified as high-sodium foods [9]. The foods with sodium content between 120 and 600 mg/100 g were classified as medium-sodium foods. These are shown as green, yellow, and red according to a horizontal bar chart to indicate the percentage of low-sodium, medium-sodium, and high-sodium foods. 

The proportion of pre-packaged foods meeting and not meeting WHO global sodium benchmarks and the difference in sodium content of the corresponding products was calculated and presented as median and IQR. The analyses were conducted using IBM SPSS V.26.0.

## 3. Results

### 3.1. The Sodium Content for Chinese Pre-Packaged Foods

From a total of 7825 pre-packaged foods collected from China, 38 were excluded due to incorrect or incomplete nutrition information, 44 were excluded due to missing ingredient information, and 83 were excluded due to not being classified into target categories (Figure 2). Finally, this study included a total of 7660 products. Chinese pre-packaged foods were classified into 20 categories and 38 subcategories (Table 1).

The highest median sodium levels per 100 g were among sauces, dips, and dressings (3896 mg/100 g, IQR: 2059 to 6523 mg/100 g), convenience foods (1578 mg/100 g, IQR: 798 to 2073 mg/100 g), processed fish products (1470 mg/100 g, IQR: 994 to 1800 mg/100 g), processed meat products (1323 mg/100 g, IQR: 990 to 1681 mg/100 g), processed poultry products (1240 mg/100 g, IQR: 980 to 1573 mg/100 g), snack foods (750 mg/100 g, IQR: 386 to 1346 mg/100 g), processed egg products (741 mg/100 g, IQR: 618 to 907 mg/100 g), and fine dried noodles (602 mg/100 g, IQR: 250 to 658 mg/100 g). Cake and pastry products (179 mg/100 g, IQR: 60 to 280 mg/100 g), biscuits (280 mg/100 g, IQR: 185 to 414 mg/100 g), cheese (363 mg/100 g, IQR: 243 to 515 mg/100 g), frozen rice and flour products (310 mg/100 g, IQR: 54 to 435 mg/100 g), and bread and bakery products (250 mg/100 g, IQR: 205 to 320 mg/100 g) were next in sodium content. Categories with relatively low sodium content included beverages (24 mg/100 g, IQR: 13 to 43 mg/100 g), edible ices (72 mg/100 g, IQR: 54 to 86 mg/100 g), yogurt, sour milk, and similar foods (60 mg/100 g, IQR: 50 to 69 mg/100 g), cereals (67 mg/100 g, IQR: 8 to 150 mg/100 g), congee (50 mg/100 g, IQR: 22 to 512 mg/100 g), and edible oil (0 mg/100 g, IQR: 0 to 0 mg/100 g) (Table 1). 

It can be seen from Table 1 that sodium content varied significantly within certain food subcategories: cake and pastry products (47 mg/100 g in moon cakes to 238 mg/100 g in western-style pastries), biscuits (107 mg/100 g in egg roll to 480 mg/100 g in soda biscuits), snack foods (59 mg/100 g in dried fruit to 2590 mg/100 g in extruded flavouring noodles), processed meat products (458 mg/100 g in prepared meat products to 1987 mg/100 g in cured meat products), processed poultry products (693 mg/100 g in prepared poultry products to 1536 mg/100 g in dried poultry products), processed fish products (701 mg/100 g in other fish products to 1561 mg/100 g in cooked fish and seafood products), and sauces, dips, and dressings (2000 mg/100 g in pickled vegetables to 6600 mg/100 g in soy sauces). Among the subcategories of bread and bakery products, the difference in sodium content was relatively small.

### 3.2. Proportion of Chinese Pre-Packaged Foods Meeting Low-Sodium, Medium-Sodium, and High-Sodium Content

The highest proportion of high-sodium foods included products within the categories of sauces, dips, and dressings (96.6%), processed meat products (95.6%), processed poultry products (94.8%), processed fish products (91.3%), convenience foods (86.7%), processed egg products (77.3%), snack foods (60.0%), and fine dried noodles (50.3%). Bread and bakery products, cheese, biscuits, cake and pastry products, and frozen rice and flour products had a relatively high proportion of medium-sodium products. Chocolate, sugar confectionery and jelly, beverages, edible ices, yogurt, sour milk, and similar foods, and edible oil had relatively low sodium content, ranging from 0 mg/100 g to 72 mg/100 g (Figure 3).

### 3.3. The Median Sodium Content of Chinese Pre-Packaged Foods Meeting and Not Meeting WHO Global Sodium Benchmarks 

The sodium content of Chinese pre-packaged foods in each food subcategory was compared with the related sodium benchmarks (Figure 4). A large proportion of products in the subcategories of frozen rice and flour products, convenience foods, processed meat products, processed poultry products, processed fish products, and sauces, dips, and dressings did not meet the related sodium benchmarks, ranging from 55.4% to 100.0%. The proportion of foods in subcategories such as biscuits, bread and bakery products, cheese, cereals, and congee meeting sodium benchmarks was relatively high. Among these categories, the proportions of soda biscuits, sandwich biscuits, and cereals meeting the sodium benchmarks were 95.0%, 93.0%, and 91.5%, respectively.

Among foods that met sodium benchmarks and those that did not, the category with the greatest difference in median sodium content is congee (34 mg/100 g in meeting to 1243 mg/100 g in not meeting), and the difference in sodium content of products that did not meet the requirements is about 36.6 times that of products that met the requirements, followed by paste and like products (20.5 times) and frozen rice and flour products (8.8 times). The difference in median sodium content between soda biscuits that met sodium benchmarks and those that did not is minimal. The difference in median sodium content between other subcategories ranges from 1.5 to 8.4 times. (Table 2).

## 4. Discussion

High sodium intake is the major dietary risk factor for deaths and DALYs in China [26,27,28]; pre-packaged foods are gradually becoming an important source of dietary sodium for the Chinese. This study analyzed the sodium content of pre-packaged foods collected by NINH, China CDC in 20 provinces of China from 2017 to 2022. We also analyzed the proportion of foods that meet or exceed the low-sodium, medium-sodium, and high-sodium. Compared with the published articles, this study also calculated the proportion of foods that meet and do not meet the WHO global sodium benchmarks, as well as the difference in sodium content of the corresponding foods. This study provided data to support the implementation of comprehensive salt reduction strategies, such as setting salt reduction targets for different categories of pre-packaged foods, improving food reformulation, and promoting the development of FOPL.

In this study, we observed that sauces, dips, and dressings had the highest median sodium content (3896 mg/100 g), and the distribution of sodium content in this category is very wide, ranging from a minimum sodium content of 0 mg/100 g to a maximum sodium content of 23,696 mg/100 g. The sodium content of sauces, dips, and dressings in this study is much lower than in 14 Latin American and Caribbean countries (7778 mg/100 g) [29] but much higher than in countries such as the UK (440 mg/100 g) [30], the USA (600 mg/100 g) [30], and Fiji (1422 mg/100 g) [31]. Sauces, dips, and dressings are special products that are not only an ingredient in pre-packaged foods but also an important source of sodium for the Chinese [32]. In the sauces, dips, and dressings category, 96.6% are high-sodium foods, and soy sauces, pickled vegetables, fermented bean curd, and paste and like products exceeded the sodium benchmark by 88.3%, 99.6%, 100.0%, 91.2%, respectively. A study that evaluated the sodium content of sauces in the UK over the past 10 years found that 70.0% of products met the UK maximum salt targets [33]. The wide range of sodium content distribution in sauces, dips, and dressings, as well as the fact that a high proportion of UK products have achieved the goal of salt reduction targets, means that there is still a lot of room for China to reduce sodium content in this category.

Pre-packaged foods commonly consumed by Chinese consumers include convenience foods, processed meat and poultry products, and snack foods, among others [10,34,35]. We found that most of these categories belong to high-sodium foods, and more than 70% of processed meat products, processed poultry products, processed fish products, and convenience foods did not meet sodium benchmarks. Therefore, in response, stricter measures to reduce sodium content should be implemented in these categories. However, researchers reported that the rate of achievement of salt reduction targets in categories such as processed meat and fish products and instant noodles was very low. In China, only 7.1% of processed meat and fish products met the UK’s sodium reduction targets [36], while 26% of instant noodles met the Pacific Salt Reduction Target (1600 mg/100 g) and 24% met the South Africa 2016 Target (1500 mg/100 g) [37]. Reducing the sodium content of pre-packaged foods that are high in sodium, high in consumption, or both can help reduce nearly 90% of the sodium intake from pre-packaged foods [38]. Therefore, strong sodium reduction policies should be implemented to regulate the sodium content of these categories, and technical issues should not become an obstacle to the high sodium content of these categories. There is evidence to support the feasibility of new approaches to reducing sodium content in pre-packaged foods, such as salt removal, salt replacement, flavour modification, functional modification, or physical modification [39].

In 2021, the WHO established global sodium benchmarks for 18 categories and 97 subcategories that were based on the lowest value for each subcategory from existing national and regional targets [25]. These sodium benchmarks ensure that products from all countries contain the same amount of sodium and provide a reference point for countries to set targets for salt in foods. If all pre-packaged foods met WHO global sodium benchmarks, the average sodium intake of the Chinese population would be reduced by 13.9% [38]. However, achieving this goal means that approximately 46.9% of Chinese packaged food needs to be reformulated [38]; this result is similar to the study by Martini et al. [40], in which they found the sodium content of most of the items of cereal-based products currently sold on the Italian market is much higher than sodium benchmarks. These results indicate that further efforts are needed globally to reduce the sodium content in pre-packaged foods. It is noteworthy that China has implemented voluntary salt reduction targets for pre-packaged foods, labelled “Healthy Choice” FOPL on lower-sodium products, and held an annual “China Salt Reduction Week” to help Chinese reduce salt intake.

Comprehensive salt reduction strategies in countries such as Finland and the UK have shown significant results. The multicomponent salt-reduction initiative led by the Finnish Government [41,42] resulted in a decrease in the average daily salt intake in Finland by 36% [43]. The comprehensive salt-reduction strategy led by the UK Government has reduced the sodium content of a wide range of packaged food categories by 20% to 50%, and overall population salt intake has decreased by 15% [15]. Comprehensive strategies could generally achieve the biggest reductions in salt consumption across an entire population; food reformulation may be one of the best strategies for reducing sodium intake [44,45]. The current Chinese adult’s average salt intake is 86% higher than the WHO’s recommendation [8]. This means that China needs to take stronger measures to reduce sodium intake and monitor and evaluate the effectiveness of these measures.

The latest report from the WHO stated that by accelerating the implementation of salt reduction policies, it is possible to meet the 2013 target of a 20% reduction in sodium intake and potentially avoid over 7 million deaths by 2030 [7]. China is committed to implementing and updating salt reduction measures, including developing FOPWL, revising salt reduction targets for pre-packaged foods, and promoting the use of low-sodium salt in China. Currently, NINH, China CDC has established a working group on guidelines for salt reduction in collaboration with the Chinese Nutrition Society and has revised Guidelines for Salt Reduction in the Chinese Food Industry. This Guideline provides a detailed classification of pre-packaged foods based on food characteristics, processing techniques, recipes, and Chinese eating habits, and plans to set the latest average salt content target and maximum salt content target for sodium reduction in pre-packaged foods. This Guideline also provides food manufacturers with scientific and practical guidance on salt reduction technologies, sets nutritional food processing standards and work targets, creates a progressive nutritional culture and policy environment, and guarantees support for consumer health. It is important to note that the National Nutrition Plan (2017–2030) aims to reduce daily sodium intake by 20%, and the WHO aims to reduce sodium intake by 30% by 2030.

The strength of this study is the fact that it included a large number of pre-packaged foods in the Chinese food and beverage supply. Compared with the published articles, this study also calculated the proportion of foods that met and did not meet the WHO global sodium benchmarks, as well as the difference in sodium content of the corresponding foods. An important limitation of the current research was that it was based on the nutritional information provided on the nutrition label and did not include actual measurements of sodium in pre-packaged foods.

## 5. Conclusions

This study analyzed the sodium content of 7660 products in 20 categories and 38 subcategories. The highest median sodium levels per 100 g were among sauces, dips, and dressings, convenience foods, processed fish products, processed meat products, processed poultry products, snack foods, processed egg products, and fine dried noodles, and these were classified as high-sodium foods. A large number of pre-packaged foods currently collected in China have a sodium content above the global sodium benchmarks. This study provided data to support the implementation of comprehensive salt reduction strategies, such as setting salt reduction targets for different categories of pre-packaged foods, improving food reformulation, and promoting the development of FOPL to help reduce sodium intake among the Chinese population and further reduce diet-related NCDs.

## Figures and Tables

**Figure 1 nutrients-15-04862-f001:**
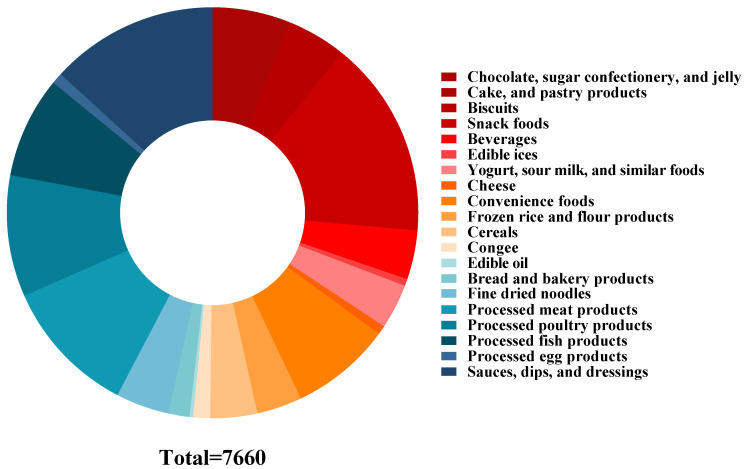
Distribution of Chinese pre-packaged foods.

**Figure 2 nutrients-15-04862-f002:**
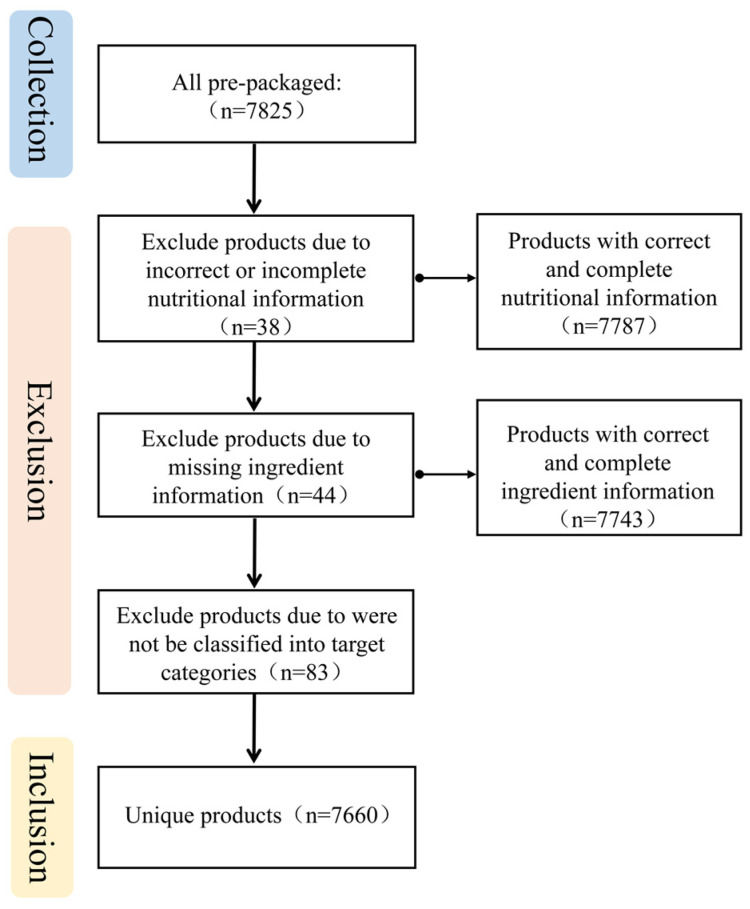
Chinese pre-packaged foods selection process.

**Figure 3 nutrients-15-04862-f003:**
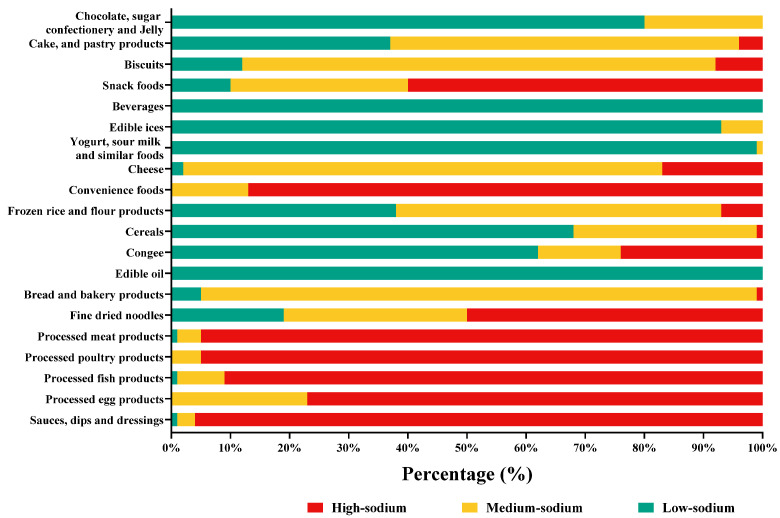
Proportion of Chinese pre-packaged foods meeting low-sodium, medium-sodium, and high-sodium content by categories.

**Figure 4 nutrients-15-04862-f004:**
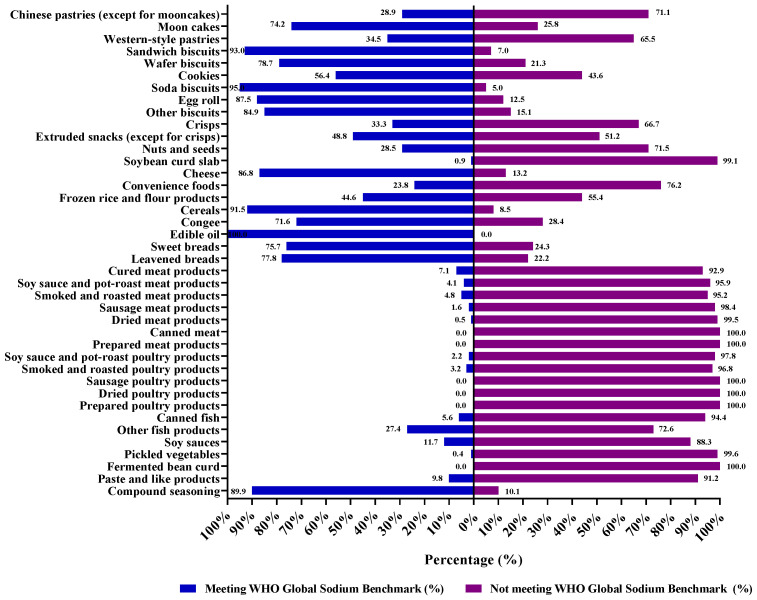
The percentage of Chinese pre-packaged foods meeting WHO global sodium benchmarks. Reference global sodium benchmarks in [25]. Cookies/sweet biscuits—265 mg/100 g; Pies and pastries—120 mg/100 g; Crackers/savoury biscuits—600 mg/100 g; Nuts, seeds, and kernels—280 mg/100 g; Potato, vegetable, and grain chips—500 mg/100 g; Extruded snacks—520 mg/100 g; Highly processed breakfast cereals—280 mg/100 g; Processed cheese—720 mg/100 g; Pasta, noodles, and rice or grains with sauce or seasoned (dry-mix, concentrated)—770 mg/100 g; Ready-to-eat meals composed of a combination of carbohydrate and either vegetable or meat, or all three combined—250 mg/100 g; Soups (ready-to-serve, canned, and refrigerated soups)—235 mg/100 g; Salted butter, butter blends, margarine, and oil-based spreads—400 mg/100 g; Sweet and raisin breads—310 mg/100 g; Leavened bread—330 mg/100 g; Canned fish—360 mg/100 g; Processed fish and seafood products, raw—270 mg/100 g; Raw meat products and preparations—230 mg/100 g; Comminuted meat products, heat treated (cooked)—540 mg/100 g; Comminuted meat products, non-heat preservation—280 mg/100 g; Potato, vegetable and grain chips—500 mg/100 g; Extruded snacks—830 mg/100 g; Pickled vegetables—550 mg/100 g; Tofu and tempeh—280 mg/100 g; Bouillon and soup stock (concentrated)—15,000 mg/100 g; Soy sauce and fish sauce—4840 mg/100 g; Other Asian-style sauces—680 mg/100 g.

**Table 1 nutrients-15-04862-t001:** The sodium content of Chinese pre-packaged foods by categories.

Number	Food Category	n	Sodium Content (mg/100 g)		
			Median	IQR	Range
1	Chocolate, sugar confectionery, and jelly	154	53	23–102	0–600
2	Cake and pastry products	316	179	60–280	0–851
2a	Chinese pastries (except for moon cakes)	135	203	101–307	0–851
2b	Moon cakes	97	47	25–121	0–744
2c	Western-style pastries	84	238	187–321	53–545
3	Biscuits	359	280	185–414	0–956
3a	Sandwich biscuits	71	337	220–400	27–810
3b	Wafer biscuits	47	155	80–240	6–492
3c	Cookies	39	260	192–324	103–497
3d	Soda biscuits	20	480	417–495	278–660
3e	Egg roll	16	107	69–188	8–451
3f	Other biscuits	166	300	233–464	0–956
4	Snack foods	1191	750	386–1346	0–6120
4a	Crisps	99	572	400–727	107–1104
4b	Extruded snacks (except for crisps)	283	530	270–798	0–4826
4c	Nuts and seeds	256	456	232–756	0–3912
4d	Extruded flavouring noodles	223	2590	2221–2748	776–3698
4e	Preserves	68	345	20–1368	0–6120
4f	Dried fruit	46	59	16–163	0–933
4g	Soybean curd slab	216	1004	827–1325	117–2608
5	Beverages	297	24	13–43	0–110
6	Edible ices	41	72	54–86	6–257
7	Yogurt, sour milk, and similar foods	272	60	50–69	32–190
8	Cheese	53	363	243–515	115–1600
9	Convenience foods	608	1578	798–2073	104–6852
10	Frozen rice and flour products	271	310	54–435	0–866
11	Cereals	283	67	8–150	0–800
12	Congee	102	50	22–512	0–2730
13	Edible oil	21	0	0–0	0–0
14	Bread and bakery products	127	250	205–320	95–606
14a	Sweet breads	37	249	186–313	95–546
14b	Leavened breads	90	251	212–325	101–606
15	Fine dried noodles	322	602	250–658	0–2100
16	Processed meat products	822	1323	990–1681	64–4718
16a	Cured meat products	28	1987	1478–2432	700–4718
16b	Soy sauce and pot-roast meat products	241	1250	973–1589	89–3160
16c	Smoked and roasted meat products	21	1022	930–1735	494–1967
16d	Sausage meat products	61	1015	873–1200	484–1967
16e	Dried meat products	398	1500	1238–1792	64–3684
16f	Canned meat	51	800	702–858	587–1300
16g	Prepared meat products	22	458	306–687	269–1170
17	Processed poultry products	732	1240	980–1573	254–3094
17a	Soy sauce and pot-roast poultry products	581	1296	1046–1620	254–3094
17b	Smoked and roasted poultry products	62	1252	894–1528	301–2898
17c	Sausage poultry products	67	985	857–1100	546–1340
17d	Dried poultry products	5	1536	1245–1580	1180–1600
17e	Prepared poultry products	17	693	544–940	386–1529
18	Processed fish products	611	1470	994–1800	39–6160
18a	Cooked fish and seafood products	477	1561	1160–1860	406–6160
18b	Canned fish	72	1116	787–1410	168–4000
18c	Other fish products	62	701	238–884	39–1948
19	Processed egg products	66	741	618–907	194–2706
20	Sauces, dips, and dressings	1012	3896	2059–6523	0–23,696
20a	Soy sauces	231	6600	5907–7333	3140–10,120
20b	Fermented bean curd	69	3525	2975–3932	2080–5859
20c	Pickled vegetables	255	2000	1655–2360	520–12,646
20d	Paste and like products	82	6416	4045–8007	0–12,000
20e	Compound seasoning	375	3896	1690–5447	120–23,696

**Table 2 nutrients-15-04862-t002:** The sodium content of Chinese pre-packaged foods meeting and not meeting WHO global sodium benchmarks, median (IQR).

Food Category	Meeting Global Sodium Benchmarks			Not Meeting Global Sodium Benchmarks		
	n	Sodium (mg/100 g)		n	Sodium (mg/100 g)	
		Median	IQR		Median	IQR
Chinese pastries (except for moon cakes)	39	56	32–86	96	280	189–464
Moon cakes	72	34	19–60	25	184	138–274
Western-style pastries	29	168	130–193	55	300	240–354
Sandwich biscuits	66	311	218–380	5	768	658–789
Wafer biscuits	37	135	66–196	10	412	317–471
Cookies	22	199	160–224	17	325	283–373
Soda biscuits	19	480	396–480	1	660	660–660
Egg roll	14	102	53–168	2	364	320–407
Other biscuits	141	276	209–365	25	699	633–747
Crisps	33	360	265–408	66	681	572–754
Extruded snacks (except for crisps)	138	267	190–409	145	769	645–919
Nuts and seeds	73	138	16–197	183	600	433–836
Soybean curd slab	2	174	145–202	214	1005	843–1331
Cheese	46	321	239–420	7	972	800–1024
Convenience foods	145	530	357–733	463	1849	1383–2188
Frozen rice and flour products	121	48	14–89	150	422	377–506
Cereals	259	40	7–121	24	321	280–413
Congee	73	34	11–54	29	1243	684–1909
Edible oil	21	0	0–0	NA	NA	NA
Sweet breads	28	224	165–256	9	400	326–499
Leavened breads	70	238	204–261	20	383	362–411
Cured meat products	2	742	721–762	26	2032	1590–2485
Soy sauce and pot-roast meat products	10	396	100–422	231	1266	1010–1594
Smoked and roasted meat products	1	752	752–752	20	1237	971–1735
Sausage meat products	1	484	484–484	60	1043	882–1200
Dried meat products	2	178	121–234	396	1500	1239–1792
Canned meat	NA	NA	NA	51	800	702–858
Prepared meat products	NA	NA	NA	22	458	306–687
Soy sauce and pot-roast poultry products	13	510	471–532	568	1304	1062–1645
Smoked and roasted poultry products	2	414	357–470	60	1254	900–1549
Sausage poultry products	NA	NA	NA	67	985	857–1100
Dried poultry products	NA	NA	NA	5	1536	1245–1580
Prepared poultry products	NA	NA	NA	17	693	544–940
Canned fish	4	320	206–341	68	1135	793–1479
Other fish products	17	124	53–194	45	796	619–1016
Soy sauces	27	4520	4520–4793	204	6754	6422–7433
Pickled vegetables	1	520	520–520	254	2000	1657–2360
Fermented bean curd	NA	NA	NA	69	3525	2975–3932
Paste and like products	8	335	0–412	74	6855	4479–8517
Compound seasoning	337	3456	1460–4720	38	19,200	17,652–20,000

Notes: NA—not applicable—means all items in these categories either met or did not meet WHO global sodium benchmarks.

## Data Availability

Data sets generated during the study are available from the corresponding author on reasonable request.

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
