# Peer review of "Sodium Content in Pre-Packaged Foods in China: A Food Label Analysis"

_nutrients, 2023, doi:10.3390/nu15234862_

Round 1

Reviewer 1 Report

Comments and Suggestions for Authors

The authors analyzed sodium Content in Pre-packaged Foods in China.

The paper is clear, relevant for the field, and presented in a well-structured manner. All figures/tables/images/schemes are appropriate, easy to interpret and understand and properly show the data. Conclusions are consistent, clear and supported by the results and all possible limitations are clearly pointed out

The cited references are mostly recent publications 

The study has significant importance for the general public as well the scientific.

The importance of the study would certainly be increased by analyzing the sodium content of certain foods for the purpose of checking declarations as authors also clearly stated as a study limitation but this does not affect the quality of the study itself.

Regarding inaccuracies within the text, there are some typos (for example in Introduction section, line 35 "High sodium intake leading cause of high blood pressure[1, 2], cardiovascular dis- 35 ease[3], stroke[4], stomach cancer[5], and premature mortality[6]." there is missing the verb "is"..)

Author Response

Manuscript title: “Sodium Content in Pre-packaged Foods in China: A Food Label
Analysis”, Manuscript ID nutrients-2734699
Response to Reviewer 1 Comments
Dear editor and reviewers
We would like to thank you for your thoughtful comments. As you are concermed, there are
several problems that need to be addressed. According to your nice suggestions, we have
made extensive corrections to our resubmitted manuscript, the detailed corrections are listed
below. The reviewer comments are laid out below in normal font and specific concerns have
been numbered. Our response is given in bold font and changes to the revised manuscript are
highlighted in yellow.
Point 1: Regarding inaccuracies within the text, there are some typos (for example in
Introduction section, line 35 "High sodium intake leading cause of high blood pressure[1, 2],
cardiovascular dis- 35 ease[3], stroke[4], stomach cancer[5], and premature mortality[6]."
there is missing the verb "is"..).
Response 1: Thanks for your careful checks, we have corrected the typos “High sodium
intake is an important risk factor of high blood pressure[1, 2], cardiovascular disease[3],
stroke[4], stomach cancer[5], and premature mortality[6]”. (Line 35, page 1) .
Thanks very much for taking your time to review this manuscript. Your careful review has
helped to make our study clearer and more comprehensive.
Best regards
Dr. Chao Gao
Corresponding author
Key Laboratory of Trace Element Nutrition of National Health Commission, National Institute for
Nutrition and Health, Chinese Center for Disease Control and Prevention, 29 Nanwei Road, Beijing
100050, China
Email:[email protected];
Tel: +8618210789809

Reviewer 2 Report

Comments and Suggestions for Authors

Dear Authors:

Regarding the manuscript with title “Sodium Content in Pre-packaged Foods in China: A Food Label Analysis”, I have some minor comments to address.

Comment 1:

Authors must change “would be analyzed” by “were analyzed”

Comment 2:

Lines 24-28: Authors must change “The highest median sodium content were among sauces, dips and dressings (3896mg/100 g), convenience foods (1578 mg/100 g), processed fish products (1470 mg/100 g), processed meat products (1323 mg/100 g), processed poultry products (1240 mg/100 g), snack foods (750 mg/100 g), processed egg products (741 mg/100 g), and fine dried noodles (602 mg/100 g), belong to high-sodium foods”

by

“High sodium foods include sauces, dips and dressings (3896mg/100 g), convenience foods (1578 mg/100 g), processed fish products (1470 mg/100 g), processed meat products (1323 mg/100 g), processed poultry products (1240 mg/100 g), snack foods (750 mg/100 g), processed egg products (741 mg/100 g), and fine dried noodles (602 mg/100 g)

Comment 3:

Line 23: Authors must change “these foods also” by “these foods was also”

Comment 4:

Line 40: The value of 9.3g/day is 86% higher when compared with 5g/day. Thus I kindly ask authors to change the sentence “which is 54% higher” by “which is 86% higher”

Comment 5:

Line 40: Regarding the sentence “China is experiencing a nutrition transition”, authors referred only to the increase of pre packaged foods consumption?

Comment 6:

Line 44: regarding the sentence “93% of the 96”, I think authors wanted to say “93 of the 96”. I kindly ask authors to confirm this issue.

Comment 7:

Lines 70-74: This study provides data to support the implementation of comprehensive salt reduction strategies, such as setting salt reduction targets for different categories of pre-packaged foods, improving food reformulation, and promoting the development of FOPL, to help reduce sodium intake among the Chinese population and further reduce diet-related noncommunicable diseases (NCDs).

I suggest authors to change this sentence to the final of Discussion.

Comment 8:

On Methods, authors have to clarify in which reference it was based the classification of the foods on 20 different categories?

Comment 9:

Line 137: Authors must change “A total of 7825” by “From a total of 7825”

Comment 10:

Lines 142-148: The highest median sodium levels per 100 g were among sauces, dips and dressings (3896mg/100 g, IQR: 2059 to 6523 mg/100 g), convenience foods (1578 mg/100 g, IQR: 798 to 2033 mg/100 g), processed fish products (1470 mg/100 g, IQR: 994 to 1800 mg/100 g), processed meat products (1323 mg/100 g, IQR: 990 to 1681 mg/100 g), processed poultry products (1240 mg/100 g, IQR: 980 to 1573 mg/100 g), snack foods (750 mg/100 g, IQR: 386 to 1346 mg/100 g), processed egg products (741 mg/100 g, IQR: 618 to 907 mg/100 g), and fine dried noodles (602 mg/100 g, IQR: 250 to 658 mg/100 g) (Table 1).

Regarding the previous sentence, I suggest authors to also present information regarding the categories within medium and low sodium content.

Comment 11:

Line 167: I suggest authors to change “Categories with the highest proportion of high-sodium foods include” by “The highest proportion of high-sodium foods include products within the categories of”

Comment 12:

Line 178: I suggest authors to change “Pre-packaged Foods that Meeting and Not Meeting” by  “Pre-packaged Foods meeting and not meeting”

Comment 13:

On the legend of Figure 4, authors must add the reference of the reference global sodium benchmarks

Comment 14:

Line 211: I suggest authors to change “foods that meeting and not meeting” by “foods meeting and not meeting”

Comment 15:

Line 304: “that met and did not meet” by “that meet and not meet”

Comments on the Quality of English Language

Minor editing of English required. Comments were made in this regard.

Author Response

Manuscript title: “Sodium Content in Pre-packaged Foods in China: A Food Label Analysis”, Manuscript ID nutrients-2734699

Response to Reviewer 2 Comments

Dear editor and reviewers

We would like to thank you for your thoughtful comments. As you are concermed, there are several problems that need to be addressed. According to your nice suggestions, we have made extensive corrections to our resubmitted manuscript, the detailed corrections are listed below. The reviewer comments are laid out below in normal font and specific concerns have been numbered. our response is given in bold font and changes to the revised manuscript are  highlighted in yellow.

Point 1: Authors must change “would be analyzed” by “were analyzed”

Response 1: Thanks for your careful checks. As suggested by the reviewer, in our resubmitted manuscript, we have corrected “And the proportion of pre-packaged foods that meet or exceed the low-sodium, medium-sodium, and high-sodium were analyzed”. (Line 22, page 1), (Line 68, page 2)

Point 2: Lines 24-28: Authors must change “The highest median sodium content were among sauces, dips and dressings (3896mg/100 g), convenience foods (1578 mg/100 g), processed fish products (1470 mg/100 g), processed meat products (1323 mg/100 g), processed poultry products (1240 mg/100 g), snack foods (750 mg/100 g), processed egg products (741 mg/100 g), and fine dried noodles (602 mg/100 g), belong to high-sodium foods”

by

“High sodium foods include sauces, dips and dressings (3896mg/100 g), convenience foods (1578 mg/100 g), processed fish products (1470 mg/100 g), processed meat products (1323 mg/100 g), processed poultry products (1240 mg/100 g), snack foods (750 mg/100 g), processed egg products (741 mg/100 g), and fine dried noodles (602 mg/100 g)

Response 2: Thanks for your careful checks. As suggested by the reviewer, in our resubmitted manuscript, we have corrected ”High sodium foods include sauces, dips and dressings (3896mg/100 g), convenience foods (1578 mg/100 g), processed fish products (1470 mg/100 g), processed meat products (1323 mg/100 g), processed poultry products (1240 mg/100 g), snack foods (750 mg/100 g), processed egg products (741 mg/100 g), and fine dried noodles (602 mg/100 g).”(Lines 24-27, page 1)

Point 3: Line 23: Authors must change “these foods also” by “these foods was also”

Response 3: Thanks for your careful checks Based on your comments, we have corrected these foods also into these foods was also.(Lines 23-24, page 1)

Point 4: Line 40: The value of 9.3g/day is 86% higher when compared with 5g/day. Thus I kindly ask authors to change the sentence “which is 54% higher” by “which is 86% higher”

Response 4: Thanks for your careful checks. We have corrected which is 54% higher into which is 86% higher(Lines 40, page 1)

Point 5: Line 40: Regarding the sentence “China is experiencing a nutrition transition”, authors referred only to the increase of pre packaged foods consumption?

Response 5: At present, there is relatively little literature on the contribution of Chinese pre-packaged foods to sodium levels in China. However, we have learned from the literature that the consumption of pre-packaged foods in China has increased significantly compared to the 1990s. Based on your comments, we have deleteed “China is experiencing a nutrition transition”. And our research provides data to support the provision of data on the consumption of packaged foods in China.

References: The food retail revolution in China and its association with diet and health (10.1016/j.foodpol.2015.07.001); Nutrition transition and chronic diseases in China (19902019): industrially processed and animal calories rather than nutrients and total calories as potential determinants of the health impact (10.1017/S1368980021003311)(Lines 40-41, page 1)

Point 6: regarding the sentence “93% of the 96”, I think authors wanted to say “93 of the 96”. I kindly ask authors to confirm this issue.

Response 6: Thanks for your careful checks Based on your comments, we have corrected “In 96 national salt reduction initiatives, 89 countries combined 2 or more implementation strategies.” References 16: A Systematic Review of Salt Reduction Initiatives Around the World: A Midterm Evaluation of Progress Towards the 2025 Global Non-Communicable Diseases Salt Reduction Target (https://doi.org/10.1093/advances/nmab008)(Lines 42-44, page 1)

Point 7: Lines 70-74: This study provides data to support the implementation of comprehensive salt reduction strategies, such as setting salt reduction targets for different categories of pre-packaged foods, improving food reformulation, and promoting the development of FOPL, to help reduce sodium intake among the Chinese population and further reduce diet-related noncommunicable diseases (NCDs). I suggest authors to change this sentence to the final of Discussion.

Response 7: Based on your comments, we have changed this sentence to the final of Discussion.(Lines 331-336, page 11)

Point 8: On Methods, authors have to clarify in which reference it was based the classification of the foods on 20 different categories?

Response 8: According to the Standard on Nutrition Labelling of Pre-packaged Foods (GB 28050-2011), the Standards for Uses of Food Additive (GB 2760-2014), Regulation of Food Composition Data Expression (WS/T 464-2015), as well as various national standards, group standards, and industry standards for food published by Chinese goverment , with full consideration of the characteristics of pre-packaged foods, their processing technologies, formulation, and Chinese dietary habits, they have been classified step by step. On line 111, we pointed out that the specific standards for classifying each food are detailed in Supplementary Table S1. (Lines 110, page 3)

Point 9: Line 137: Authors must change “A total of 7825” by “From a total of 7825”

Response 9: Thanks for your careful checks. As suggested by the reviewer, we have corrected A total of 7825 into From a total of 7825. (Lines 138, page 3)

Point 10: Lines 142-148: The highest median sodium levels per 100 g were among sauces, dips and dressings (3896mg/100 g, IQR: 2059 to 6523 mg/100 g), convenience foods (1578 mg/100 g, IQR: 798 to 2033 mg/100 g), processed fish products (1470 mg/100 g, IQR: 994 to 1800 mg/100 g), processed meat products (1323 mg/100 g, IQR: 990 to 1681 mg/100 g), processed poultry products (1240 mg/100 g, IQR: 980 to 1573 mg/100 g), snack foods (750 mg/100 g, IQR: 386 to 1346 mg/100 g), processed egg products (741 mg/100 g, IQR: 618 to 907 mg/100 g), and fine dried noodles (602 mg/100 g, IQR: 250 to 658 mg/100 g) (Table 1).

Regarding the previous sentence, I suggest authors to also present information regarding the categories within medium and low sodium content.

Response 10: Thanks for your careful checks. We have present information regarding the categories within medium and low sodium content. ”Cake, and pastry products (179 mg/100 g, IQR: 60 to 280 mg/100 g), biscuits (280 mg/100 g, IQR: 185 to 414 mg/100 g), cheese (363 mg/100 g, IQR: 243 to 515 mg/100 g), frozen rice and flour products (310 mg/100 g, IQR: 54 to 435 mg/100 g), bread and bakery products (250 mg/100 g, IQR: 205 to 320 mg/100 g) were next in sodium content. Categories with relatively low sodium content include beverages (24 mg/100 g, IQR: 13 to 43 mg/100 g), edible ices (72 mg/100 g, IQR: 54 to 86 mg/100 g), yogurt, sour milk and similar foods (60 mg/100 g, IQR: 50 to 69 mg/100 g), cereals (67 mg/100 g, IQR: 8 to 150 mg/100 g), congee (50 mg/100 g, IQR: 22 to 512 mg/100 g), and edible oil (0 mg/100 g, IQR: 0 to 0 mg/100 g)”.(Lines 149-157, page 4)

Point 11: Line 167: I suggest authors to change “Categories with the highest proportion of high-sodium foods include” by “The highest proportion of high-sodium foods include products within the categories of”

Response 11: Thanks for your careful checks. As suggested by the reviewer, we have corrected Categories with the highest proportion of high-sodium foods include into The highest proportion of high-sodium foods include products within the categories of(Lines 177-178, page 6)

Point 12: Line 178: I suggest authors to change “Pre-packaged Foods that Meeting and Not Meeting” by “Pre-packaged Foods meeting and not meeting”

Response 12: Thanks for your careful checks. According to the requirements of the journal, the title of the results section must be capitalised. (Lines189-190, page 7)

Point 13: On the legend of Figure 4, authors must add the reference of the reference global sodium benchmarks.

Response 13: Thanks for your careful checks. Based on your comments, we have add the reference of the reference global sodium benchmarks. (Lines 212, page 8)

Point 14: Line 211: I suggest authors to change “foods that meeting and not meeting” by “foods meeting and not meeting”

Response 14: Thanks for your careful checks. As suggested by the reviewer, we have corrected foods that meeting and not meeting into foods meeting and not meeting. (Lines 224-225, page 8)

Point 15: Line 304: “that met and did not meet” by “that meet and not meet”

Response 15: Thanks for your careful checks. As suggested by the reviewer, we have corrected that met and did not meet into that meet and not meet. (Lines 234, page 9)

Thanks very much for taking your time to review this manuscript. Your careful review has helped to make our study clearer and more comprehensive.

Best regards

Dr. Chao Gao

Reviewer 3 Report

Comments and Suggestions for Authors

This manuscript offers detailed and well-contextualized information about the health impact of sodium intake.

The introduction provides mortality data that seem at odds with the 2023 WHO report that shows quite different numbers for sodium-related mortality

(https://iris.who.int/bitstream/handle/10665/366393/9789240069985-eng.pdf?sequence=1).

Also, it seems odd for the introduction to state that China has more than half of global sodium-related mortality. That implies a much higher proportationate effect within China than in the rest of the human planetary population. It would seem advisable to double-check the reported mortality numbers.

The authors do a commendable job of discussing the context for their findings, but the broader impact of the manuscript would benefit from a more explicit summary of the implications of their results for policy and practice. Some of those implications are scattered throughout the manuscript, and compiling a concentrated discussion of policy and practice would add to the footprint of the research.

Comments on the Quality of English Language

Only minor editing seems to be needed.

Author Response

Manuscript title: “Sodium Content in Pre-packaged Foods in China: A Food Label Analysis”, Manuscript ID nutrients-2734699

Response to Reviewer 2 Comments

Dear editor and reviewers

We would like to thank you for your thoughtful comments. As you are concermed, there are several problems that need to be addressed. According to your nice suggestions, we have made extensive corrections to our resubmitted manuscript, the detailed corrections are listed below. The reviewer comments are laid out below in normal font and specific concerns have been numbered. our response is given in bold font and changes to the revised manuscript are  highlighted in yellow.

Point 1: The introduction provides mortality data that seem at odds with the 2023 WHO report that shows quite different numbers for sodium-related mortality.

Response 1: Thanks for your careful checks. As suggested by the reviewer, We used data from the report of 2023 WHO GLOBAL REPORT ON SODIUM INTAKE REDUCTION (https://iris.who.int/bitstream/handle/10665/366393/9789240069985-eng.pdf?sequence=1) for sodium-related mortality -1.89 million (Line 37, page 1). In our resubmitted manuscript, we have corrected High sodium intake led to 1.9 million deaths globally. 

Point 2: Also, it seems odd for the introduction to state that China has more than half of global sodium-related mortality. That implies a much higher proportationate effect within China than in the rest of the human planetary population.

Response 2: Thanks for your careful checks. Excessive sodium intake led to 1.7 million deaths in China[7], and 1.9 million deaths globally[8]. Reference 7 is from Figure 2: Number of deaths and percentage of DALYs related to the leading level 3 risk factors in China in 2017 of the study on Mortality, morbidity, and risk factors in China and its provinces, 1990–2017: a systematic analysis for the Global Burden of Disease Study 2017 (http://dx.doi.org/10.1016/S0140-6736(19)30427-1). Reference 8 is from Impact of individual components of diet on mortality in the results section of the study on Health effects of dietary risks in 195 countries, 1990–2017: a systematic analysis forthe Global Burden of Disease Study 2017 (http://dx.doi.org/10.1016/S0140-6736(19)30041-8). In our resubmitted manuscript, we delete the sentence that excessive sodium intake led to 1.7 million deaths in China .(Line 37, page 1)

Thanks very much for taking your time to review this manuscript. Your careful review has helped to make our study clearer and more comprehensive. 

Best regards

Dr. Chao Gao

Key Laboratory of Trace Element Nutrition of National Health Commission, National Institute for Nutrition and Health, Chinese Center for Disease Control and Prevention, 29 Nanwei Road, Beijing 100050, China

Email:[email protected];

Tel: +8618210789809
